# Hydrogen Sulfide Produced by Gut Bacteria May Induce Parkinson’s Disease

**DOI:** 10.3390/cells11060978

**Published:** 2022-03-12

**Authors:** Kari Erik Murros

**Affiliations:** Institute of Clinical Medicine, University of Eastern Finland (UEF), 70211 Kuopio, Finland; ext-murrokar@hus.fi

**Keywords:** hydrogen sulfide, Parkinson’s disease, gut microbiome, *Desulfovibrio*, enteroendocrine cells, cytochrome c, alpha-synuclein, iron, reactive oxygen species, influenza

## Abstract

Several bacterial species can generate hydrogen sulfide (H_2_S). Study evidence favors the view that the microbiome of the colon harbors increased amounts of H_2_S producing bacteria in Parkinson’s disease. Additionally, H_2_S can easily penetrate cell membranes and enter the cell interior. In the cells, excessive amounts of H_2_S can potentially release cytochrome c protein from the mitochondria, increase the iron content of the cytosolic iron pool, and increase the amount of reactive oxygen species. These events can lead to the formation of alpha-synuclein oligomers and fibrils in cells containing the alpha-synuclein protein. In addition, bacterially produced H_2_S can interfere with the body urate metabolism and affect the blood erythrocytes and lymphocytes. Gut bacteria responsible for increased H_2_S production, especially the mucus-associated species of the bacterial genera belonging to the Desulfovibrionaceae and Enterobacteriaceae families, are likely play a role in the pathogenesis of Parkinson’s disease. Special attention should be devoted to changes not only in the colonic but also in the duodenal microbiome composition with regard to the pathogenesis of Parkinson’s disease. Influenza infections may increase the risk of Parkinson’s disease by causing the overgrowth of H_2_S-producing bacteria both in the colon and duodenum.

## 1. Introduction

In humans, hydrogen sulfide (H_2_S) plays various roles in a myriad of physiologic processes relating to inflammatory, immune, endocrine, respiratory, vascular and neuromodulatory actions [1]. Furthermore, H_2_S is endogenously produced in the human cells by enzymes including cystathionine β-synthase (CBS) and cystathionine γ-lyase (CSE), which both use L-cysteine as a substrate, and via 3-mercapto-sulfurtransferase (3-MST) pathway that uses 3-mercaptopyruvate as a substrate. Both the brain and colonic tissue generate and modulate H_2_S production via the activity of CSE and CBS [2]. In healthy humans, the plasma baseline levels of H_2_S have been reported to lie in the range of 34 μM–274 μM [3,4]. In the gut lumen, the sulfate-reducing bacteria (SRB) and the bacteria of the desulfhydrase enzyme are notable H_2_S producers. In addition, various bacteria which are homologs of the mammalian CBS, CSE, and 3-MST enzymes are capable of producing H_2_S. At low concentrations, bacterially produced H_2_S displays cytoprotective properties by maintaining gut mucus integrity but is toxic to the host at high concentrations [5]. When administered to the human body, H_2_S diffuses rapidly to blood and, after administration, only a small part of H_2_S remains in a soluble form [6]. In the blood, H_2_S combines easily with hemoglobin (Hb) and methemoglobin (metHb) [7,8]. MetHb and H_2_S form relatively stable metHb-H_2_S complexes which have a slow reduction rate. Thus, metHb keeps Hb in an oxygen-binding form and acts as a scavenger and regulator of sulfide in the blood [8,9]. In the cells, the main route of H_2_S elimination is mitochondria, by a chain of several oxidative enzymes [10]. Overall, the toxicity-dose relationship in H_2_S exposition has been documented to be extremely steep and toxic symptoms such as hypotension and apnea occur when the concentration of free H_2_S reaches only a few moles per litre in the blood [6]. High H_2_S concentrations are toxic to cells, causing the inhibition of the cytochrome oxidase, a hemeprotein which is the last enzyme of the electron transport chain in the mitochondria [11]. In addition, H_2_S induces a release of cytochrome c protein from the mitochondrial membrane, an event though to be associated with the etiopathogenesis of Parkinson’s disease (PD) [12,13].

## 2. H_2_S Releases Cytochrome c from the Mitochondria—Start for Alpha-Synuclein Aggregation

Cytochrome c is (Cyt c) is a small heme protein responsible for the electron shuttle between Complex III and Complex IV of the respiratory chain in the mitochondria. It binds to the mitochondrial inner membrane partly by weak readily mobilized electrostatic interactions [14]. In an experiment where cultured human lung fibroblasts were treated with increased H_2_S concentrations, a release of Cyt c from the mitochondria was observed [15]. In a study where human gingival epithelial cells were incubated for up to three days with air containing a low concentration (50 ng/mL) of H_2_S, a remarkable release of Cyt c from the mitochondria was noted [16]. Accordingly, when human dental pulp stem cells were exposed to air somposed of a similar low concentration of H_2_S, significant increases in the apoptotic enzyme levels (caspase-9 and -3), accompanied by increases in the Cyt c release from the mitochondria were observed [17]. Special information on the dynamics of Cyt c release from the mitochondria provides information on the effects of rotenone, a poisonous isoflavone which has been used in animal studies to induce experimental PD. In a proportional rotenone titration, a proportional increase in the release of Cyt c from the mitochondria was documented in this study [18]. Of importance, neurons can tolerate Cyt c release into the cytosol without inducing apoptosis. Although Cyt c concentrations were already modestly increased in the cytosol, the activity of the apoptosis indicator caspase 9 was not increased [18]. In its compact state, Cyt c possesses peroxidase activity [19]. In the preapoptotic phase when Cyt c relase has already started, Cyt c can interact with anionic lipids and due to these interactions, Cyt c can develop remarkable peroxidase activity [20,21]. The anionic lipids containing Cyt c-peroxidase can utilize cytoplasmic alpha-synuclein (aSyn) as a peroxidase substrate which leads to an aggregation of aSyn and Cyt c, a reaction reflecting mostly a protective role of aSyn against apoptosis [21]. The native Cyt c, which leads to peroxidase activity, can potentially take part in the aSyn oligomerization and aggregation in the cytosol. It has been shown that aSyn aggregation and aSyn radical formation occurs in the interaction between Cyt c, aSyn and hydrogen peroxide, a member of the reactive oxygen species (ROS) [22,23]. Alterations in the iron metabolism induced by H_2_S probably take part in the aSyn aggregation. It has been reported that sulfide releases iron from the mammalian ferritin and rises the ferrous iron levels of the cytosolic labile iron pool (LIP) and some evidence favors the view that H_2_S can reduce intracellular bound ferric iron to form unbound ferrous iron [24,25,26]. Furthermore, it has been demonstrated that ferrous iron promotes both aSyn aggregation and transmission by inhibiting the autophagosome-lysosome fusion [27]. In case the iron content of the LIP reaches abnormally high levels, ROS formation in the cell eventually increases [26,28]. The possible presence of magnetite nanoparticles produced by some *Desulfovibrio* gut bacteria may be an additional factor in the increase in cytosolic ROS levels in the gut cells [13,29,30]. The increased ROS formation likely favors the emergence of aSyn oligomers and fibrils in the presence of Cyt c and aSyn (Figure 1)

## 3. Overgrowth of H_2_S Producing Gut Bacteria—Putative Consequences

Several bacterial species generate H_2_S in the human gastrointestinal canal. A prominent overgrowth of these species may lead to a spectrum of undesired consequences. Notably, H_2_S is capable of penetrating cell membranes easily without any help of a facilitator for this transport [31]. To counteract the toxic effects of H_2_S the colonic mucosal cells are subject to an efficient mechanism to dispose H_2_S by oxidizing it to thiosulfates [32]. In mitochondria, a chain of enzymatic reactions starting with the actions of sulfide quinone oxidoreductase (SQR) catalyze the oxidation of sulfide to thiosulfate, which is further oxidized to sulfate and excreted by the kidney [10]. In case H_2_S is produced in abnormally high amounts by the gut bacteria, the concentration of H_2_S may exceed the capacity of the gut cells to detoxify all H_2_S and part of it may end up in the blood. Experimentally, it has been shown that free H_2_S levels are reduced in the plasma and gastrointestinal tissue of germ-free mice, indicating that the gut microbiota activity can increase plasma H_2_S concentrations [33]. A large study where plasma metabolome data were integrated with the gut microbiome data of PD patients and healthy controls revealed changes in sulfur metabolism driven by *Akkermansia muciniphila* and *Bilophila wadsworthia* bacteria [34]. Notably, the secretion potential of H_2_S was detected to increase in the PD microbiome in that study. Further support for the view that increased H_2_S production occurs in PD is a study where blood metHb content was reported to be increased in PD patients [35]. In addition, it has been shown that PD patients have significantly higher concentrations of H_2_S in the cerebrospinal fluid when compared to control subjects [36]. Studies providing information on the correlations between blood concentrations of H_2_S and quantities of gut bacteria producing H_2_S are not availabe. If the production of H_2_S by gut bacteria leads to unphysiologically high blood concentrations, the brain tissue may be especially sensitive to the toxic actions of H_2_S as the activity of SQR enzyme activity is apparently very low in the mammalian brain cells [37]. As to changes in the blood constituents, CD8+ cytotoxic T-lymphocytes have been especially reported to be decreased in the blood of PD patients and this decrease seems to be associated with the severity of PD [38]. Furthermore, H_2_S effects may provide an explanation for the finding as it has been found that exogenous H_2_S induces the cell death of peripheral blood lymphocytes with relevant subset specifity for the CD8+ T lymphocytes and natural killer cells [39]. As an additional observation, increased H_2_S levels likely modify plasma uric acid (UA) concentrations which reflect the intestinal metabolism of dietary purines such as xanthine [40]. Additionally, H_2_S has been reported to decrease UA formation from xanthine, giving one explanation for the finding that plasma UA levels are lowered in patients with PD [41,42].

## 4. H_2_S and aSyn Containing Gut Cells

Of special concern is the part of the gut epithelial cells which faces the gut lumen without vascular covering. Consequently, those epithelial cells such as the enteroendocrine cells (EECs) which lack the protective role of blood Hb and metHb to scavenge H_2_S are at an increased risk of H_2_S toxicity. An excess production of H_2_S by bacterial species belonging to the genus *Desulfovibrio* (*gDSV*) has been suggested to induce Cyt c release from the mitochondria and aSyn oligomerization in the EECs [12,13]. It has been reported that aSyn is expressed in EECs both of the small and large intestine and that EECs are connected to enteric nerves [43,44]. In addition, EECs express both pre- and postsynaptic proteins, implying that EECs both send and receive neural signals [44]. Similarly to EECs, enteric neurons carry a potential risk to produce aSyn oligomers when exposed to high levels of H_2_S as enteric neurons contain aSyn [45]. Hypothetically, if aSyn oligomers are produced in EECs and enteric neurons, part of them are possibly secreted in the blood. Significantly elevated levels of oligomeric forms of aSyn have been observed in the blood plasma of PD patients [46]. However, the origin of these oligomeric aggregates has remained an open issue. As a potential mechanism, aSyn oligomers and fibrils propagate, in the same way as prions, from the EECs and enteric neurons by a cell-to-cell mechanism to the lower brain stem via the vagal nerve (Figure 1). Experimentally, it has been shown that different forms of aSyn, after being injected into the intestinal wall, are transported to the brainstem via the vagal nerve [47]. In another study on transgenic rats expressing an excess of human aSyn, injections of aSyn fibrils into the duodenum wall induced aSyn pathology both in the parasympathetic and sympathetic pathways [48]. 

## 5. H_2_S Producing Colonic Bacteria and PD

With regard to humans, the excess production of H_2_S by gut bacteria may induce the emergence of PD. Sulfate-reducing bacteria (SRB) form the main group of H_2_S-producing bacteria in the feces of healthy people [49]. In this group, the most prevalent genera are the hydrogen- and lactate utilizing *gDSV*, acetate utilizing genus *Desulfobacter,* and hydrogen and propionate utilizing genus *Desulfobulbus* [50,51]. Notably, SRB are the only gut microbes that rely on inorganic sulfate for energy conservation [51]. Numerous other bacteria that do not belong to the SRB group produce H_2_S as well. Among others, bacterial genera such as *Alistipes*, *Bacteroides*, *Escherichia*, *Enterobacter*, *Clostridium*, *Collinsella*, *Fusobacterium*, *Klebsiella*, *Oscillibacter*, *Prevotella*, *Proteus*, *Porphyromonas*, *Streptococcus*, *Veillonella*, and *Yearsinia* include species that have the capacity for primary H_2_S production via cysteine degradation [51,52]. Of special interest, *Bilophila wadsworthia* bacteria produces H_2_S via the taurine degradation pathway [51]. As to specific bacterial species known to produce H_2_S, *Helicobacter pylori*, *Clostridium difficile*, and *Desulfovibrio desufuricans* have been reported to be associated with the occurrence of PD [13,53,54]. 

Numerous studies have been conducted on the microbiome changes in PD with variable results. A large meta-analysis of ten case–control studies on the gut microbiome in PD showed that the fecal samples of PD patients were most consistently enriched by the genera *Lactobacillus*, *Bifidobacterium* and *Akkermansia* [55]. In contrast, the butyrate-producing genera *Roseburia*, *Blautia*, *Faecalibacterium*, *Moryella* and *Anaerostipes* were found to be less enriched. In a large microbiome-wide association study (MWAS) the genera *Prevotella*, *Corynebacterium_1*, and *Porphyromonas*, as a cluster of co-occurring opportunistic pathogens, were reported to be enriched in PD [56]. As noted, the genera *Prevotella* and *Porphyromonas* are H_2_S producers. In addition, sulfite reductase, an iron flavoprotein enzyme which produces H_2_S, has been suggested to be present in the species of the genus *Corynebacterium* [57]. However, concerning the enrichment of the genus *Prevotella* in the fecal samples of PD patients in the MWAS study, contrasting results have been reported in some other studies [58,59,60]. In the MWAS study, the relative abundances of the genera *Butyricoccus*, *Lachnospira*, *Fusikatenibacter*, *Roseburia*, *Blautia*, *Agathobacter* and *Faecalibacterium*, all of which are butyrate producers, were found to be significantly decreased in the PD gut microbiome. An overgrowth of H_2_S-producing bacteria provides a reasonable explanation for the reduced population of butyrate-reducing bacteria. Furthermore, H_2_S inhibits acetyl-CoA synthesis and produces CoA-persulfide when reacting with CoA, an essential molecule in the butyrate synthesis pathways [61,62]. In an innovative study on a synthetic microbiome community, composed of several butyrate-producing bacteria, it was observed that H_2_S, in concentrations similar to those produced by *Desulfovibrio piger*, inhibited butyrate production across the designed bacterial community, including the members *Faecalibacterium prausnitzii* and and *Roseburia intestinalis* [63]. Decreased butyrate production most probably reflects the amount of butyrate-producing gut bacteria. Advocating this view, it has been reported that the amount of butyrate correlate with the abundancies of bacteria of the genus *Blautia* and bacterial species such as *Faecalibacterium prausnitzii* and *Roseburia faecis* [64]. An enrichment of the genera *Lactobacillus* and *Bifidobacterium* in the fecal samples of PD patients has been reported in several studies [55,65]. The finding can be explained, at least partly, by the availability of these probiotics as commercial products. 

Concerning SRB such as *gDSV*, their role in the gut microbiota of PD patients requires particular attention. In an Italian study, many bacterial genera, including *gDSV,* were observed to be enriched in the fecal samples of PD patients. However, after adjusting for several covariates, only the genus *Veillonella* remained as the enriched genus [66]. The finding is of interest as *Veillonella* species produce H_2_S from L-cysteine and an enrichment of this genus has been found to be enriched in the fecal microbiome of the PD patients in an earlier study [52,58,67]. In two Chinese studies, the relative abundance of the family Desulfovibrionaceae was observed to be significantly increased in the fecal samples of PD patients [58,68]. In addition, in a study from southern China, the relative abundance of *gDSV* was increased in PD patients [68]. Further confirmation for the overgrowth of the family Desulfovibrionaceae in the gut microbiota of PD patients is provided in a Canadian study where the Desulfovibrionaceae and Christensenellaceae families showed overgrowth in the fecal samples of PD patients [69]. The family Christensenellaceae has been reported to be significantly increased in the PD microbiota [55]. One explanation for its increased abundance is linked to body mass index (BMI). Patients with PD have significantly lower BMIs when compared to healthy controls and on the other hand, an inverse correlation exists between BMI and the relative abundance of the Christensenellaceae family [70,71].

In contrast to a focus on the relative abundances of a multitude of gut bacteria representing various taxonomic levels, one study provided data on the absolute concentrations of the *gDSV* species *Desulfovibrio desulfuricans*, *D. fairfieldensis*, *D. piger* and *D. vulgaris* in the fecal samples of PD patients and healthy controls [13]. Excluding *D. vulgaris*, not found in the PD samples, the *gDSV* species were present at significantly higher concentrations in the PD samples than in the control samples. In addition, the presence of the periplasmic [FeFe]-hydrogenase, which was used as a common denominator of different *gDSV* species, correlated with the presence of PD [13]. Apparent cross-feeding takes place between *gDSV* and genus *Akkermansia*. In studies on fecal samples, the relative overgrowth of the genus *Akkermansia* is a common finding in PD patients [55,56,65]. *Akkermansia muciniphila*, a bacterium residing abundantly in the mucus layer of the large intestine releases sulfate in mucin fermentation, thus offering sulfate to the mucosa-associated SRB [72,73]. A study on sequence analyses of functional gene clones of colonic biopsy samples revealed that SRB populations associate with the mucosa throughout the colon and are phylogenetically related to hydrogenotrophic *Desulfovibrio piger*, *Desulfovibrio desulfuricans* and *Bilophila wadsworthia* bacteria [74]. In colonic mucus samples of healthy people, *Bilophila wadsworthia* was reported to have a high colonisation rate [75]. Of apparent importance, the relative abundance of the genus *Bilophila* has been reported to be significantly increased in the fecal samples of PD patients when compared to controls, and to correlate with the clinical stage of PD by the Hoehn and Yahr clinical criteria [76]. As an apparent weakness, the gut microbiome studies on PD have focused mainly on the relative abundancies of various bacterial taxa occurring in the fecal samples. The composition of the mucosal-associated microbiome evidently differs significantly from that of the fecal microbiome, as demonstrated in an analysis comprising both sigmoidoscopy and fecal samples [77]. Genus *Escherichia* resides especially in the mucosal area [77,78]. Notably, a significantly more intense staining for *Eschericia coli* has been reported in the sigmoid mucosal samples of PD patients than in the samples of the control subjects [79]. In agreement with this study, the relative abundance of the genus *Escherichia-Shigella* has been reported to be significantly increased in the fecal samples of PD patient and to correlate with the severity and duration of PD [80].

## 6. H_2_S Producing Small Intestinal Bacteria and PD

Apart from the changes in the colonic microbiota composition, changes in the quantities of small intestinal bacteria likely relate to the pathogenesis of PD as it suggests the high prevalence of small intestinal bacterial overgrowth (SIBO) in the PD population. According to a large meta-analysis, based on lactulose-hydrogen breath test results, a strong association was reported to exist between SIBO and PD [81]. Although increased hydrogen production from the small intestinal microbiome seems to take place in several PD subjects, it is not yet known which bacterial genera or species in the small intestine explain this increase. Hydrogen-utilizing bacteria such as *Desulfovibrio* species and *Bilophila wadsworthia* likely benefit from these circumstances. At a general level, the small intestinal microbiome composition has been found to differ markedly from the fecal microbiome composition in humans [82]. A study performed on SIBO and non-SIBO subjects provides interesting information about the duodenal microbiome changes in the SIBO subjects at a general level [82]. In that study, over four-fold higher relative abundance of the class Gammaproteobacteria and three-fold higher relative abundance of the class Deltaproteobacteria was reported to be present in the duodenal aspirates of SIBO subjects when compared to non-SIBO subjects. Of further interest, the family Enterobacteriaceae represented 89 % of the total relative abundance of the Gammaproteobacteria in the duodenum of the SIBO subjects [83]. Notably, hydrogen sulfide-producing genera such as *Escherichia*, *Klebsiella*, and *Proteus* belong to the family Enterobacteriaceae. As to the family members of the class Deltaproteobacteria, especially concerning the family Desulfovibrionaceae, they are best known for their ability to produce H_2_S. Swallowed bacteria from the oral cavity, emerging, e.g., from the areas of periodontal infections, may possibly play a role in modulating the small intestinal microbiome composition. In a large-scale cohort study, periodontitis was shown to increase the risk of PD [84]. Subgingival deposits occurring with periodontal disease, including cases with aggressive periodontitis, have been reported to contain SRB (which mainly represent the class Deltaproteobacteria) and, among others, bacterial genera such as *Porphyromonas* and *Prevotella* [85,86]. Of interest, in a Finnish study on oral mucosal biofilm, significantly increased abundances of the *Prevotella* and *Veillonella* genera were observed in PD patients [87].

## 7. Viral Infections, PD, and H_2_S Producing Gut Bacteria

Influenza infection evidently increases the risk of PD. In a population-based case–control study in Canada, a significant association was reported to exist between a history of severe influenza and PD [88]. Recently, a case–control study based on the data of over 60 000 individuals from the Danish National Patient Registry showed that a history of influenza was significantly associated with a later occurrence of PD [89]. In addition, the Spanish flu (influenza A subtype H1N1) has been considered to be a risk factor for a later PD occurrence. As to animal studies, it has been shown that influenza A virus (H5N1) can induce a disruption of the mucus layer integrity of the small intestine and cause enteric dysbiosis by increasing the relative amounts of Gammaproteobacteria and Bacilli [90]. In a mice study, the influenza A (PR8) virus infection was demonstrated to induce a significant increase in the intestinal *Escherichia coli* species [91]. As to another study, influenza A (PR8) infection caused a significant increase in the Enterobacteriaceae population in the stool samples of the wild-type mice [92]. In addition, influenza A virus H3N2 infection has been reported to increase the fecal content of the genus *Escherichia* in mice [93]. As to human studies, an interesting cross-sectional study on alterations in the gut microbiota of 30 patients with COVID-19 infection, 24 patients with influenza (H1N1), and 30 matched healthy controls showed that the relative abundance of the genus *Escherichia-Shigella* was significantly higher in the fecal samples of H1N1 patients than in controls [94]. 

Hepatitis C virus (HCV) infection has been reported to be a significant risk factor for Parkinson’s disease [95]. HCV infection can be classified into three types, which consist of the persistently normal serum alanine transferase type, chronic hepatitis, and liver cirrhosis. In all these stages, an enrichment of the family Enterobacteriaceae present in the gut microbiome has been a consistent finding [96]. Evidently, influenza and hepatitis C virus infections may be inductors for the development of PD by inducing gut dysbacteriosis, whereby an overgrowth of the H_2_S-producing members of the family Enterobacteriaceae, especially the genus *Escherichia*, play a crucial role. Clinical studies on fecal microbiome content in PD provide support for the view that the Enterobacteriaceae family is an important player in the pathogenesis of PD. Notably, this family has been reported to be significantly more abundant in the fecal samples of PD patients when compared to healthy controls [80,97,98]. Postural instability and gait difficulty have been reported to correlate with the relative abundance of the Enterobacteriaceae in the fecal samples of PD patients [97]. In addition, an increase in Enterobacteriaceae in the fecal samples has been reported to be associated with the non-tremor dominant subtype of PD [99]. 

## 8. Bacterially Produced H_2_S and Risk Factors for PD

Advancing age has been established as the largest risk factor for developing PD. Globally, the prevalence of PD begins rise steeply in the age range of 60–70 years and peaks in the age range of 80–90 years [100]. As to the colonic microbiome in advancing age, a community study on changes in the microbiome composition during the aging process showed that in older people, a sharp and continuous increase takes place in the relative abundance of the H_2_S-producing *Desulfovibrio*, *Bilophila* and *Corynebacterium* gut bacteria [101]. In case additional and substantial overgrowth of the H_2_S producing bacterial genera such as *Escherichia* or *gDSV* takes place, the risk for PD development will likely increase as a function of age. 

In addition to aging, male gender is an established risk factor for PD. Furthemore, with regard to the incidence rates, male to female ratios have been reported to vary from 1.37 to 3.7 [102]. Estrogen, especially 17beta-estradiol which has been reported to display neuroprotective properties may provide an explanation for this difference [103,104]. It has been stated that long-term exposure to estrogen may be important in the PD risk reduction [104]. Experimental studies on ischemia models have shown that 17beta-estradiol prevents the release of Cyt c from mitochondria and by this action, estradiol displays cytoprotective properties [105,106]. Ultimately, given that H_2_S levels increase in the gut cells and perhaps even at the brain level, estadiol can probably counteract H_2_S actions by preventing the release of Cyt c from the mitochondrial membrane. 

## 9. Conclusions

Considerable amounts of evidence support the view that an overgrowth of H_2_S-producing bacteria takes place in the gut microbiome of PD patients. It is plausible that by inhibiting acetyl-CoA synthesis, H_2_S decreases the amount of butyrate-producing gut bacteria. Furthemore, H_2_S can diffuse easily to gut cells and the vascular compartment. In blood circulation, some part of the bacterially produced H_2_S may end up at the brain level. When the gut cells are exposed to unphysiologically high concentrations of H_2_S, the release of Cyt c from the mitochondria will likely begin. In addition, H_2_S increases the iron content in the labile iron pool of the cell cytosol and the ROS content. In case the cell expresses aSyn, in the way that EECs and enteric neurons do, a development of aSyn oligomers and fibrils may start in the presence of Cyt c and increased ROS. The toxic oligomers and fibrils may spread to the lower brainstem via the vagal nerve and some oligomers will likely end up in the blood circulation. As to H_2_S producers, species of the genera *Desulfovibrio*, *Escherichia*, *Bilophila*, *Porhyromonas*, *Prevotella*, *Corynebacterium*, *Veillonella*, *Helicobacter*, and *Clostridium* invite particular attention when identifying the role of different bacterial genera in the etiology of PD. The bacterial composition of the microbiome in duodenum require special attention in PD as it is likely that the quantities and characteristics of the small intestinal bacteria, including the H_2_S-producing bacteria, differ significantly from those of the colon. Notably, studies on the bacterial composition of duodenal microbiome in PD are lacking. In future, microbiological and metabolomics-based studies on duodenal aspirates combined with the corresponding analyses of fecal samples may offer key information on the pathogenesis of PD. In case H_2_S plays a substantial role in the pathogenesis of PD, several strategies are already available to counteract its actions. In 2003, Braak and his colleagues proposed that PD is caused by an intestinal pathogen [107]. H_2_S could be that ”pathogen”. 

## Figures and Tables

**Figure 1 cells-11-00978-f001:**
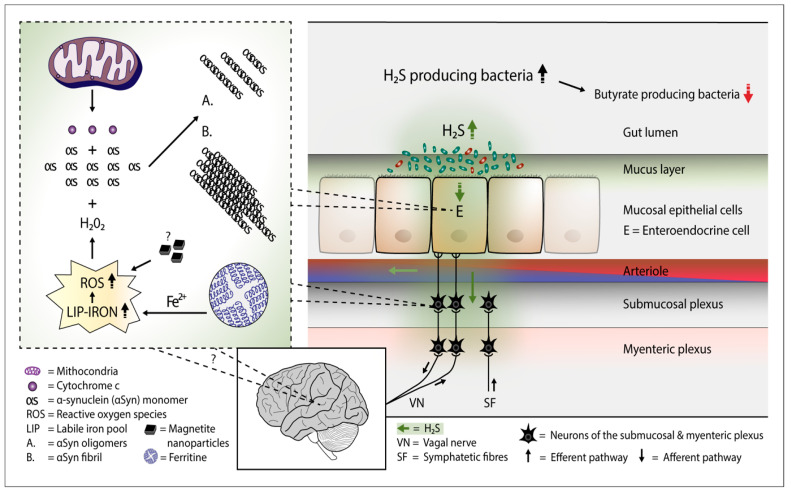
Plausible pathophysiological mechanism of Parkinson’s disease. Overgrowth of H_2_S producing gut bacteria raises H_2_S concentrations in the gut cells and blood. In the gut cells, excessively increased H_2_S releases Cyt c from the mitochondria and increases cytosolic iron (Fe^2+^) levels. Consequently, the amount of reactive oxygen species (ROS) increases. The presence of magnetite nanoparticles originating from the *Desulfovibrio* species can further increase the emergence of ROS. The co-occurence of aSyn, Cyt c, and ROS (especially hydrogen peroxide) leads to aSyn aggregation. Emerged aSyn aggregates (oligomers and fibrils) may spread in a prion-like manner to the lower brain stem via the vagal nerve. In the blood H_2_S combines with hemoglobin. Part of H_2_S may remain in a free form and possibly induce aSyn aggregation even in the brain neurons.

## Data Availability

Not applicable.

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
