# Peer review of "Hydrogen Sulfide Produced by Gut Bacteria May Induce Parkinson’s Disease"

_cells, 2022, doi:10.3390/cells11060978_

Round 1

Reviewer 1 Report

This is a well written and comprehensive review that hypothesizes the
potential role of colonic H2S-producing bacteria in the pathogenesis
of Parkinson disease. The author also proposes a potential mechanism
of H2S molecule in the aggregation of aSyn. I recommend that the
manuscript can be accepted

There is a minor comment:

1. Section 3, Line 125: The author should mention if other authors
have studied the correlation between high H2S levels and increased
H2S-producing bacteria in PD 

Author Response

Thank you for the good feedback I got !, I will  add the following sentence to the paragraph 3 (starting from the line 123, after the sentence carrying the reference number 36) :                                

For today, studies providing information on the correlations between blood concentrations of H2S and quantities of gut bacteria producing H2S are not available.         

Reviewer 2 Report

This is  a comprehensive review supporting a hypothesis that hydrogen sulfite producing bacteria could hav a major role in Parkinsons Disease

I would suggest to start with the evedence that the microbiome both colonic and small bowel have been found to be different in PD

There are different forms of PD Some have a worse prognosis is there any link with miccrobiom 

pthesisCould the conclusion state what further work is required to proove hy

Author Response

First, I like to thank the reviewer for the good rating scores and important considerations in order to improve the quality of the manuscript !

1)

As to the suggestion to possibly start the article by focusing on the the colonic and small bowel microbiome changes, I would like to say that was an optional approach when I planned to write this review. However, by focusing first on the hydrogen sulfide, I decided to especially highlight the possible role of hydrogen sulfide in the pathogenesis of PD.

However, to underline the possible role of duodenal microbiomea bit more, I made some changes to the last two sentences of the abstract (lines 17-19) The modified sentences will be now in the form as follows:

Special attention should be devoted to changes not only in the colonic but also in the duodenal microbiome composition with regards to the pathogenesis of Parkinson’s disease.  Influenza infections may increase the risk of Parkinson’s disease by causing overgrowth of H2S producing bacteria both in the colon and duodenum.

2)

The question concerning the link between the microbiome composition and prognosis of PD

with regards to different clinical PD subtypes/forms is relevant. As to the family Enterobacteriaceae, there is some information on its relation to the different forms of PD. This information has been now added to end of the paragraph number 7. A new reference (Vascellari et al. 2021, now reference number 100) has been added to the reference list.

The additional text will be as follows:

Postural instability and gait difficulty have been reported to correlate to the

relative abundance of the Enterobacteriacea in the fecal samples of PD patients [97]. In addition, an increase of Enterobacteriaceae in the fecal samples has been reported to associate with the non-tremor dominant subtype of PD [100].

3)

As to the Conclusions paragraph, concerning the further studies on this subject (?), I will add two sentences to this part (starting from the line 347, before the sentence ”In case H2S plays a substantial role…..!

The text will be as follows:      

Notably, studies on the bacterial composition of the duodenal microbiome in PD are lacking. For the future, microbiological and metabolomics-based studies on duodenal aspirates combined with similar studies on fecal samples may offer key information on the pathogenesis of PD. In case……

Reviewer 3 Report

Re: Manuscript ID: cells-1586420

I congratulate the author on his well written review dealing with the intriguing role of hydrogen sulfide produced by gut bacteria in the pathogenesis of Parkinson’s disease. The different arguments are well balanced and explored. The author has expertise in this field. Some minor changes are suggested to improve the paper.

Minor points

Line 51. Replace “Parkinsons” with “Parkinson’s”.

Line 90. Desulfovibrio in italic (Desulfovibrio).

Line 101. Replace “spred” with “spread”.

Line 104. Replace “neurons. .” with “neurons.”.

Line 105. Replace “consequencies” with “consequences”.

Line 168. Delete “bacterial”.

Line 263. Replace “utilisizing” with “utilizing”.

Paragraph 6. Bacterial strains in italic. References in bold.

Line 272. Replace “89 %” with “89%”.

Line 277. Replace “H2S” with “H2S”.

Line 283. Replace “finnish” with “Finnish”.

Line 291. Replace “Parkinson’s disease” with “PD”.

Line 295. Delete the second “the”.

Line 297. Replace “of” with “in”.

Line 307. Replace “Parkinson’s disease” with “PD”.

Line 339. Replace “amount” with “amounts”.

Lines 358-359. Replace “Hydrogen sulfide could be that pathogen” with “H2S could be the dangerous product of that pathogen”.

Line 394. Reference 12. Replace “2020” with “2021”.

Author Response

I am greatly thankful for the very positive feedback !  I am glad to correct all those typographical errors. In fact correcting these details will really enhance the quality of the article. I am also glad of the remark that bacterial strains should be written in italics. I will correct all those errors in the manuscript and use the established way by writing the names of bacterial genera and species in italics but use normal letters in case of upper taxonomic levels.  With cordial regards Kari Murros